# The influence of maternal blood glucose during pregnancy on weight outcomes at birth and preschool age in offspring exposed to hyperglycemia first detected during pregnancy, in a South African cohort

Tawanda Chivese[1]*, Magret C. Haynes[2], Hetta van Zyl[3], Una Kyriacos[2], Naomi S. Levitt[4], Shane A. Norris[5,6]

1 Department of population medicine, College of medicine, QU Health, Qatar University, Doha, Qatar,
2 Division of Nursing and Midwifery, School of Health and Rehabilitation Sciences, Faculty of Health Sciences, University of Cape Town, Cape Town, South Africa, 3 Department of Medicine, Faculty of Health Sciences, University of Cape Town, Cape Town, South Africa, 4 Chronic Disease Initiative for Africa, Department of Medicine, Faculty of Health Sciences, University of Cape Town, Cape Town, South Africa,
5 SAMRC/Wits Developmental Pathways for Health Research Unit, Department of Paediatrics, Faculty of Health Sciences, University of the Witwatersrand, Johannesburg, South Africa, 6 Global Health Research Institute, School of Human Development and Health, University of Southampton, Southampton, United Kingdom

* tchivese@qu.edu.qa

## Abstract

### Introduction

Little is known about the influence of hyperglycemia first detected in pregnancy (HFDP) on weight outcomes in exposed offspring in Africa. We investigated the influence of maternal blood glucose concentrations during pregnancy on offspring weight outcomes at birth and preschool age, in offspring exposed to HFDP, in South Africa.

### Research design and methods

Women diagnosed with HFDP had data routinely collected during the pregnancy and at delivery, at a referral hospital, and the offspring followed up at preschool age. Maternal fasting, oral glucose tolerance test 1 and 2-hour blood glucose were measured at diagnosis of HFDP and 2-hour postprandial blood glucose during the third trimester. Offspring were classified as either those exposed to diabetes first recognized in pregnancy (DIP) or gestational diabetes (GDM). At birth, neonates were classified into macrosomia, low birth weight (LBW), large for gestational age (LGA), appropriate (AGA) and small for gestational age (SGA) groups. At preschool age, offspring had height and weight measured and Z-scores for weight, height and BMI calculated.

### Results

Four hundred and forty-three neonates were included in the study at birth, with 165 exposed to DIP and 278 exposed to GDM. At birth, the prevalence of LGA, macrosomia and LBW

**Data Availability Statement:** All data are have been uploaded as Supporting information.

**Funding:** The author(s) received no specific funding for this work.

**Competing interests:** The authors have declared that no competing interests exist.

**Abbreviations:** DIP, diabetes in pregnancy; GDM, gestational diabetes mellitus; HAPO Study, Hyperglycemia and Adverse Pregnancy Outcomes Study; HbA1c, glycated haemoglobin A1C; HFDP, Hyperglycemia first detected in pregnancy; IADPSG, International Association of Diabetes and Pregnancy Study Groups; LBW, low birth weight; LGA, large-for-gestational-age; LMIC, low-to-middle-income; NICE, National Institute for Health and Care Excellence; OGTT, oral glucose tolerance test; SD, standard deviation; WHO, World Health Organization.

were 29.6%, 12.2% and 7.5%, respectively, with a higher prevalence of LGA and macrosomia in neonates exposed to DIP. At pre-school age, the combined prevalence of overweight and obesity was 26.5%. Maternal third trimester 2-hour postprandial blood glucose was significantly associated with z-scores for weight at birth and preschool age, and both SGA and LGA at birth.

## Conclusion

In offspring exposed to HFDP, there is a high prevalence of LGA and macrosomia at birth, and overweight and obesity at preschool age, with higher prevalence in those exposed to DIP, compared to GDM. Maternal blood glucose control during the pregnancy influences offspring weight at birth and preschool age.

## Introduction

The prevalence of childhood overweight and obesity is a public health concern globally, with 25% of children under the age of 5 years who are overweight or obese being in Africa [1]. While childhood overweight and obesity is plateauing in high-income countries, in Africa the prevalence in under-fives doubled to 5% during the period 2000 to 2017 [2] and the prevalence of obesity quadrupled between 1975 to 2016 [3]. Children who are overweight and obese tend to remain so in adulthood, with a consequent earlier and higher risk for cardiovascular risk factors and cardiovascular disease (CVD) [4]. As interventions during early life are more effective in reducing the risk of adulthood overweight and obesity than those in childhood [3], it is imperative to identify and intervene in children at risk of overweight and obesity.

Developmental Origins of Health and Disease (DOHaD) has shown that maternal under- and-overnutrition during pregnancy affect the offspring's future risk for overweight and obesity and consequent cardiometabolic disease [5, 6]. Offspring of women with hyperglycemia first discovered in pregnancy (HFDP) may be particularly at risk for being large-for-gestational-age (LGA) at birth and overweight and obesity during childhood, due to exposure to a high glucose uterine environment [7]. This has led the World Health Organization (WHO) Report of the Commission on Ending Childhood Obesity to emphasize the need to improve the diagnosis and management of HFDP [1], as one of the strategies to reduce risk of childhood overweight and obesity.

The WHO guidelines of 2013 define HFDP as either diabetes first recognized in pregnancy (DIP) or gestational diabetes (GDM) [8]. The term DIP is used to describe women who, at the time of testing for GDM during the second trimester, have blood glucose levels that are similar to the cut-offs that are used to diagnose diabetes outside pregnancy. However, these women are not classified as having type 2 diabetes, as there is not enough research on whether their blood glucose levels normalize or not after the pregnancy. Many older guidelines and studies have classified both DIP and GDM groups as GDM, although DIP may in some cases, imply that the foetus is likely to be exposed to hyperglycaemia for a longer period compared to GDM and consequently, untreated hyperglycaemia for a longer period until screening and treatment [9]. This is because a proportion of women with DIP may consist of women who may have had undiagnosed diabetes until GDM screening during the second trimester pregnancy [10]. In Africa, up to two thirds of people living with diabetes are undiagnosed [11], and it is possible that the proportion of women with undiagnosed diabetes before the pregnancy is higher in

women with DIP, compared to other regions. Data comparing the effect of the two subtypes of HFDP on offspring overweight and obesity during childhood are scarce [7].

Maternal glucose levels at diagnosis of HFDP are linked to neonate weight outcomes at birth. The Hyperglycemia and Adverse Pregnancy Outcomes (HAPO) Study of 23 216 participants from 10 countries, demonstrated a linear graded relationship between maternal blood glucose at diagnosis of GDM and birth size [12]. The major guidelines for the diagnosis of HFDP, including the International Association of Diabetes and Pregnancy Study Groups (IADPSG) and the WHO 2013 criteria, have since adapted their criteria, based on this landmark study. However, the lack of an African cohort in the HAPO study may limit the applicability of the findings to the continent, where a high prevalence of HFDP has been reported in several countries [13–16] and where undernutrition and overnutrition coexistence is prevalent [17]. Data from African cohorts are needed to compliment the HAPO findings.

Evidence on whether maternal blood glucose influences overweight and obesity risk during early and later childhood remains inconclusive [7]. Findings from the HAPO follow up study [18] showed no significant associations between GDM and childhood overweight and obesity at ages 10–14 years, after adjusting for maternal BMI, but, at the same age, there were significant associations with other measures of adiposity such as body fat percentage, waist circumference and sum of skinfolds. Three systematic reviews [7, 19, 20] found higher BMI z-scores in children exposed to GDM, compared to those from normoglycemic pregnancies, although in some of the included studies the association was not statistically significant once adjusted for maternal BMI. Arguably, as a child grows older, other factors such as socioeconomic factors, diet and physical activity, contribute to a child's nutrition and the influence of maternal hyperglycemia may lessen. Nevertheless, more research evidence is needed.

In South Africa, recent epidemiological studies reported prevalences of HDFP and GDM of 26% [16] and 9% [15], respectively, suggesting that up to a quarter of pregnancies may be complicated by HFDP. Since most provinces in the country use a selective risk factor screening approach and consequently may miss up to 50% of women with HFDP, a substantial proportion of children are likely to be exposed to untreated HFDP [21]. This study was undertaken to contribute to the understudied area of the cardiometabolic outcomes of children from HFDP in South Africa and elsewhere in Africa. The main aim was two-fold: (i) to investigate the influence of maternal blood glucose during pregnancy, and offspring weight at birth, weight and prevalence of overweight and obesity at ages 5–6 years in children exposed to HFDP, and (ii) to compare the prevalence of overweight and obesity at birth and at follow-up, between children exposed to DIP and those exposed to GDM, in an African cohort.

## Research design and methods

A cohort of offspring exposed to HFDP had birthweight measured at birth and was followed up at ages 5–6 years. Routine pregnancy and delivery data were collected on all mothers diagnosed and managed with HFDP at a major tertiary hospital in the Western Cape province of South Africa between 1 September 2010 to 31 August 2011. At that time, modified National Institute for Health and Care Excellence (NICE) 2008 guidelines were used to diagnose GDM [fasting blood glucose > 5.5mmol and/or oral glucose tolerance test (OGTT) 2-hour glucose > 7.8mmol/l] [22]. Five to six years after the pregnancy, a cross-sectional study [10, 23] was carried out, where each mother was recalled and assessed for diabetes and CVD risk factors. Each mother was asked to bring her offspring 5–6 from the index pregnancy. Neonates who had the following characteristics at birth were excluded from the analysis: children from multiple births, premature, congenital birth disorders, admitted into neonatal intensive care at birth and neonates hospitalized with serious conditions.

## Data collected

During the index pregnancy, maternal age, BMI and gestational age at booking, maternal HIV status, treatment for HFDP, type of birth delivery and gestational age at delivery were routinely collected by the attending clinician. Maternal glucose measures included fasting glucose, OGTT 1-hour and OGTT 2-hour glucose concentrations at HFDP diagnosis, and routine fasting and postprandial glucose measurements during the third trimester. We retrospectively classified the children into 2 groups, using a modified WHO 2013 criteria and maternal blood glucose values at HFDP diagnosis, into DIP-exposed (maternal fasting glucose at HFDP diagnosis of at least 7.0mmol/l and /or OGTT 2-hour glucose of at least 11.1 mmol/l) and GDM-exposed (maternal fasting glucose at HFDP diagnosis of at least 5.5mmol but less than 7.0 mmol/l and/or OGTT 2-hour glucose of at least 7.8 mmol/l but less than 11.1 mmol/l). Maternal fasting blood glucose and 1 and 2-hour postprandial blood glucose were measured weekly during the third trimester, as part of routine clinical monitoring, until delivery, and the mean blood glucose calculated for each woman.

## Outcomes

Birthweight was measured by the attending clinician, who was not part of the study. Gestational age at birth was calculated from the first day of the last menstrual period (LMP) and ultrasound estimation, which was carried out between 18–23 weeks of gestational age estimated from LMP, according to the guidelines of the South African Western Cape Province Department of Health [24]. Birthweight z-scores and birthweight percentiles for gestational age and gender were computed using the *International Newborn Size at Birth Standards* software [25]. Neonates with birthweight percentile<10% were classified as small-for-gestational-age (SGA) while those with birthweight> 90% were classified as large-for-gestational-age (LGA). Additionally, neonates with birthweight<2500 grams were classified as low-birth-weight (LBW) and neonates with birthweight>4000 grams as macrosomic. At follow-up, trained study staff measured the children's anthropometry in light clothing and without shoes. Height was measured eight to the nearest 0.1cm, using a wall-mounted stadiometer. Weight was measured to the nearest 0.1kg, using a calibrated digital scale. All measurements were taken in duplicate and the average calculated. Z-scores for weight, height and BMI were calculated using the WHO Child Growth Standards STATA *igrowup* package for children 5 years old or younger and the WHO Child Growth Standards STATA WHO 2007 package for children above 5 years [26]. Children over the age of 5 years with BMI z-scores above one but less than 2 were classified as overweight while children with BMI z-score of at least 2 were classified as obese. Children with BMI z-scores below -1 were classified as underweight.

## Sample size and sampling

The study sample consisted of 443 eligible neonates at birth and these were all included. The same children were eligible for follow-up and thus the sample size at follow up consisted of all children who were able to participate in the study, although there was significant attrition.

## Data analysis

We used *STATA* 15 [27] and *R statistical software* [28] for all statistical analyses, p<0.05 as a cut-off for significance and reported 95% confidence intervals (CI) for prevalence and regression estimates, where applicable. For summarizing data, we reported frequencies and proportions for categorical data, means and standard deviations (SD) for measured data, such as child anthropometry and ages, if normally distributed, and medians and interquartile ranges

(IQR) for non-normal data. We calculated the prevalence of neonates who were LGA and the prevalence of overweight and obesity at preschool age as a proportion of the children with the outcome divided by the total assessed.

We compared z-scores, LGA at birth, and overweight and obesity at preschool age between children exposed to DIP and those exposed to GDM during the pregnancy. P-values for group comparisons were computed using the chi-squared test for categorical data and the t-tests for independent groups or Wilcoxon rank-sum test (where data were not normally distributed).

Multiple variable linear regression was used to investigate the association between maternal blood glucose concentrations during pregnancy, and weight z-score at birth and preschool age and BMI at preschool age. A linear mixed-effects model was used to investigate the effect of maternal blood glucose on longitudinal offspring weight z-score, as there were no data for offspring BMI z-score at birth. Multiple variable multinomial logistic regression was used to examine the association between maternal blood glucose and the categorized outcomes of size at birth (LGA and SGA with AGA as the base outcome) and BMI category at preschool age (Overweight and Obesity with normal BMI as the base outcome). In all the models the following maternal blood glucose variables were included; 1) glucose levels at HFDP diagnosis (fasting blood glucose, OGTT 1-hour, OGTT 2-hours) and third trimester mean 2-hour postprandial blood glucose. In all models, we adjusted for maternal age at pregnancy booking, maternal BMI at pregnancy booking, gender and mode of birth delivery. In addition, size at birth (LGA or SGA vs AGA) was adjusted for in all models at preschool age. Maternal HIV was not included in the analysis due to the low prevalence in the study while fasting and 1-hour postprandial blood glucose were omitted as they had too much missing data.

The study is reported according to the Strengthening the Reporting of Observational Studies in Epidemiology (STROBE) guidelines (S1 Checklist).

## Ethics

This study protocol (S2 Doc) was approved by the Human Research Ethics Committees of the University of Cape Town (Refs: 377/2012 and 656/2015) and permission obtained to conduct research at the tertiary hospital. The study was conducted according to the ethical principles of the Helsinki Declaration [29]. At follow up, mothers gave written informed consent for their children to participate, and each child assented before they had their weight and height measured. All the children who were brought to the research site assented.

## Results

Of the 498 women treated for HFDP, 443 (89.0%) had children who were eligible for the birth-weight assessment. Of these 167 (37.7%) were followed up at ages 5–6 years. The remainder were lost to follow-up due to various reasons (Fig 1). The only differences in the baseline clinical characteristics of children followed up compared to those lost to follow up (S1 Table in S1 File) were: the lower mean gestational age at booking of the pregnancy and smaller proportion of children exposed to DIP in those seen as follow up compared to those not followed up [(15 (IQR 12–21) vs 17(IQR 13–23) weeks, and 29.9% vs 41.7% respectively, p = 0.013].

### Participant characteristics and comparison between DIP and GDM-exposed infants

Table 1 shows maternal and offspring characteristics. At booking of the pregnancy, mean maternal age was 30.5(SD 6.2) years, mean maternal BMI was 34.5(SD 8.6) kg/m2 and median gestational age was 16(IQR 12–22) weeks. Two hundred-and-twenty (49.7%) of the 443 infants were female. At HFDP diagnosis, the median fasting and OGTT 2-hour blood glucose

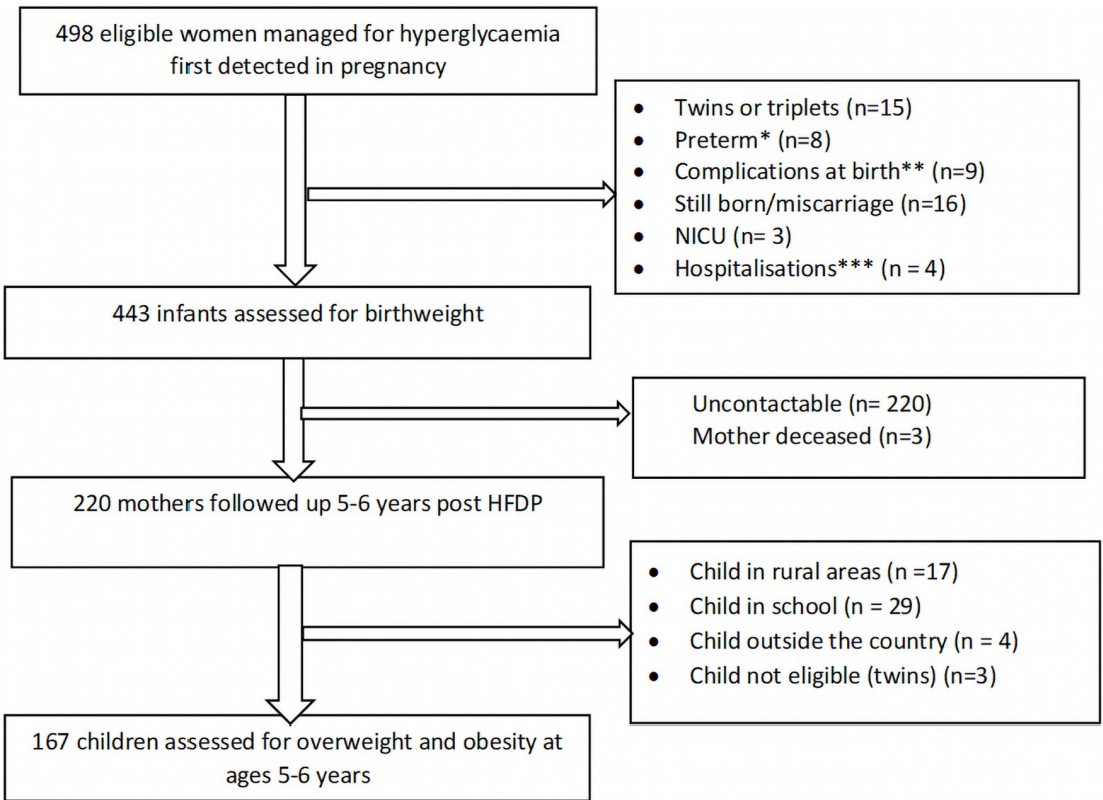

**Fig 1. Flow chart of the study.** *Preterm–extreme or very preterm: gestational age below 32 weeks (WHO classification), and/or birthweight ≤ 1500 grams. **Dandy Walker syndrome (n = 1), ventriculomegaly (n = 1), Intrauterine growth restriction (n = 4), short long bones (n = 2), kyphosis (n = 1). ***hospitalizations (multiple problems (n = 2), hypoglycemia (n = 1), hypothermia and microencephaly(n = 1)).

concentrations were 5.8 (IQR 5.2–6.6) mmol/l and 9.1 (IQR 8.3–10.6) mmol/l, respectively, with, as expected, higher concentrations in the mothers with DIP, compared to those with GDM (Table 1). The children's mean age at follow up was 5.5 (SD 0.5) years, with only 26 (15.5%) being 5 years old or younger. After classification using modified WHO 2013 criteria, 165 (37.2%) and 50 (29.9%) were classified as DIP at birth and preschool age, respectively. The maternal characteristics at follow up did not differ significantly between the DIP-exposed and GDM-exposed children (Table 1).

### Prevalence of LGA at birth and overweight and obesity at preschool age, and comparison between exposure to DIP and GDM

The median birthweight of the offspring was 3.3 (IQR 3.0–3.6) kg and their mean BMI at pre-school age was 16.12 (SD 2.92) kg/m$^2$. The DIP-exposed neonates had a significantly higher mean z-score for birthweight than GDM-exposed infants [0.52 (SD 1.20) vs 0.11 (SD 0.97), respectively, p <0.001] but there were no significant differences in weight, height and BMI, between the two groups of children at preschool age (Table 2 and Fig 2).

The prevalence of LGA at birth was 29.6% (95%CI 25.5–34.0), with a significantly higher prevalence in the neonates exposed to DIP, compared to those exposed to GDM (37.6% vs 24.8%, p = 0.018, respectively). The prevalence of macrosomia was 12.2% (95%CI 9.4%–15.6%), with a higher prevalence in DIP-exposed compared to GDM-exposed, although not

**Table 1. Characteristics of, and comparison between infants exposed to GDM and those exposed to DIP.**

| Variable | Level | Overall | DIP | GDM | p-value |
|---|---|---|---|---|---|
| **Pregnancy-related data** | | **N = 443** | **N = 165** | **N = 278** | |
| Maternal age at booking (mean (SD)) | Years | 30.5 (6.2) | 31.2 (6.0) | 30.1 (6.2) | 0.06 |
| Maternal ethnicity (n (%)) | Black | 130 (29.4) | 57 (34.6) | 73 (26.3) | 0.064 |
| | Mixed ancestry | 313 (70.7) | 108 (65.5) | 205 (73.7) | |
| Maternal BMI at booking (kg/m$^2$) (n = 385) | (Mean (SD)) | 34.5 (8.6) | 35.1 (8.3) | 34.2 (8.7) | 0.35 |
| Maternal HIV status (n (%)) | Positive | 27 (6.1) | 16 (9.8) | 11 (4.0) | 0.024 |
| Maternal CD4 count (cells/dm$^3$), n = 25 | (Median [IQR]) | 453 (364–595) | 433.0 [345.8, 465.0] | 654.0 [390.0, 874.0] | 0.089 |
| Maternal gravida | (Median [IQR]) | 3 (2–4) | 3 [2, 4] | 3 [2, 4] | 0.024 |
| Maternal parity | (Median [IQR]) | 3 (2–4) | 2 [1, 3] | 1 [0, 2] | 0.029 |
| Gestational age at booking (weeks) | (Median [IQR]) | 16 (12–22) | 18.0 [12.0, 23.0] | 15.5 [13.0, 21.0] | 0.179 |
| Fasting glucose (mmol/l) at HFDP diagnosis (n = 422) | (Median [IQR]) | 5.8 (5.2–6.6) | 7.20 [6.2, 8.5] | 5.6 [5.0, 5.9] | <0.001 |
| OGTT 2-hour glucose (mmol/l) at HFDP diagnosis (n = 381) | (Median [IQR]) | 9.1 (8.3–10.6) | 11.9 [11.1, 13.4] | 8.6 [8.1, 9.4] | <0.001 |
| Third trimester postprandial 2-hour blood glucose (n = 440) | (Median [IQR]) | 5.7 [5.1–6.3] | 5.9 [5.2–6.6] | 5.6 [5.1–6.2] | 0.031 |
| Insulin treatment for HFDP (n (%)) | Yes | 111 (25.1) | 88 (53.3) | 23 (8.3) | <0.001 |
| Oral hypoglycemics treatment (n (%)) | Yes | 141 (31.9) | 70 (42.4) | 71 (25.6) | <0.001 |
| Mode of birth delivery (n (%)) | Caesarian section | 231 (52.4) | 96 (58.2) | 135 (48.9) | 0.074 |
| | Vaginal | 210 (57.6) | 69 (41.8) | 141 (51.1) | |
| Gestational age at delivery (weeks) | Median [IQR] | 38 [38–39] | 38.0 [37.0, 38.0] | 38.0 [38.0, 39.0] | <0.001 |
| Infant gender (n (%)) | Female | 225 (50.8) | 88 (53.3) | 137 (49.3) | 0.467 |
| **Follow up data (age 5–6 years), N = 167** | | | | | |
| Child gender (n (%)) | Female | 80 (47.9) | 28 (56.0) | 52 (44.4) | 0.171 |
| Child age (years) | (Mean (SD)) | 5.5 (0.5) | 5.7 (0.5) | 5.5 (0.5) | 0.014 |
| Maternal education (n (%)) | Primary | 15 (8.9) | 8 (16.0) | 7 (6.0) | 0.093 |
| | Secondary | 129 (77.3) | 37 (74.0) | 92 (78.6) | |
| | Tertiary | 23 (13.8) | 5 (10.0) | 18 (15.4) | |
| Mother employed (n (%)) | Yes | 80 (47.9) | 23 (46.0) | 57 (48.7) | 0.878 |
| Maternal alcohol (n (%)) | Ever consumed | 72 (43.9) | 22 (44.0) | 50 (43.9) | 1.000 |
| Maternal smoking (n (%)) | Ever smoked | 61 (36.5) | 17 (34.0) | 44 (37.6) | 0.789 |

NB: n is specified where data are missing.

Abbreviations: BMI, body mass Index, FBG, fasting blood glucose, HFDP, hyperglycaemia first detected in pregnancy, OGTT, oral glucose tolerance test, DIP, diabetes in pregnancy, GDM, gestational diabetes mellitus.

significant at a 5% significance level (Table 2). At preschool age, the combined prevalence of overweight and obesity was 26.5% (95%CI 20.1–34.0), with no statistically significant differences between DIP-exposed and GDM-exposed infants (Table 2).

## Maternal blood glucose during pregnancy and birth size and overweight and obesity at preschool age

All maternal blood glucose values were consistently high for the neonates who were LGA, compared to those who were either AGA or SGA (Table 3). Postprandial blood glucose exhibited the strongest association with an increase in birth size, with the highest mean (SD) for LGA, followed by AGA and lastly SGA. Similarly, maternal fasting and OGTT 2-hour glucose at diagnosis of HFDP were both significantly higher for LGA compared to AGA but not for OGTT-1hour glucose. At preschool age, all mean maternal blood glucose parameters were high for offspring who were overweight and obese compared to those who had either normal BMI or underweight, although not statistically significant (Table 3).

**Table 2. Anthropometry at birth and preschool age, and comparison between children exposed to GDM and those exposed to DIP.**

| Variable | Level | Overall | DIP | GDM | P |
|---|---|---|---|---|---|
| **Birth** | | N = 443 | N = 165 | N = 278 | |
| Neonate birthweight (kg) | (Median [IQR]) | 3.3 [3.0–3.6] | 3.3 [3.0, 3.7] | 3.3 [3.0, 3.6] | 0.474 |
| Neonatal birthweight z-score | (Mean (SD)) | 0.26 (1.1) | 0.52 (1.20) | 0.11 (0.97) | <0.001 |
| Birthweight percentile for gestational age category (n (%)) | AGA | 291 (65.69) | 96 (58.18) | 195 (70.14) | 0.018 |
| | SGA | 21 (4.74) | 7 (4.24) | 14 (5.04) | |
| | LGA | 131 (29.57) | 62 (37.58) | 69 (24.82) | |
| Birthweight category | Macrosomia (n (%)) | 54 (12.19) | 27 (16.36) | 27 (9.71) | 0.117 |
| | LBW (n (%)) | 33 (7.45) | 12 (7.27) | 21 (7.55) | |
| **Preschool-age** | | | | | |
| Height (cm) | (Mean (SD)) | 112.50 (5.63) | 113.80 (6.68) | 112.00 (5.12) | 0.072 |
| BMI at follow up (kg/m$^2$) | (Mean (SD)) | 16.12 (2.92) | 16.57 (3.71) | 15.95 (2.55) | 0.229 |
| Child weight (kg) | (Mean (SD)) | 20.52 (4.77) | 21.59 (5.99) | 20.06 (4.09) | 0.057 |
| Child height z-score | (Mean (SD)) | -0.15 (1.02) | -0.06 (1.09) | -0.18 (0.99) | 0.503 |
| Weight z-score | (Mean (SD)) | 0.14 (1.51) | 0.28 (1.77) | 0.07 (1.39) | 0.409 |
| Child BMI z-score | (Mean (SD)) | 0.34 (1.54) | 0.35 (1.77) | 0.33 (1.46) | 0.937 |
| BMI category (n (%)), N = 155 | Underweight (z-score <-1) | 26 (16.8) | 7 (17.1) | 19 (16.7) | 0.935 |
| | Normal (-1≤z-score ≤1) | 88 (56.8) | 26 (52.0) | 78 (66.7) | |
| | Overweight (1<z-score≤2) | 24 (15.5) | 7 (17.1) | 17 (14.9) | |
| | Obese (z-score>2) | 17 (11.0) | 5 (12.2) | 12 (10.5) | |
| Combined overweight & obese category (n (%)) | BMI z-score>1 | 41 (26.5) | 12(29.3) | 29 (25.4) | 0.681 |

NB: Neonates with birthweight percentile<10% were classified as small-for-gestational-age (SGA) while those with birthweight> 90% were classified as large-for-gestational-age (LGA). Neonates with birthweight<2500 grams were classified as low-birthweight (LBW) and neonates with birthweight>4000 grams as macrosomic. Abbreviations: LGA, large for gestational age, AGA, appropriate for gestational age, SGA, small for gestational age, BMI, body mass Index, FBG, fasting blood glucose, DIP, diabetes in pregnancy, GDM, gestational diabetes mellitus.

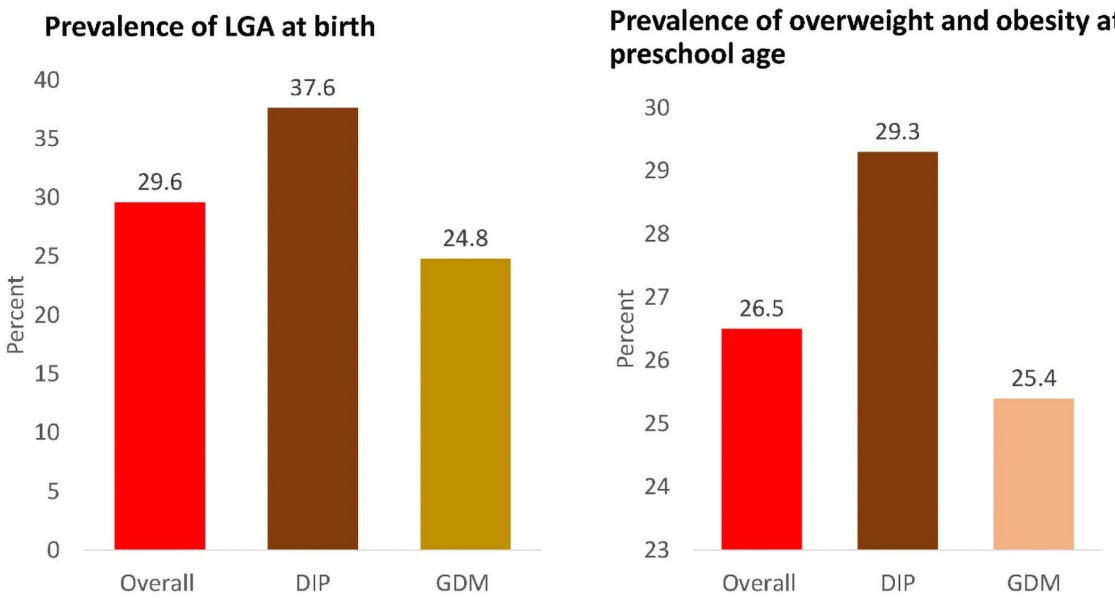

**Fig 2. Prevalence of LGA at birth and overweight and obesity at preschool age and comparison between DIP and GDM exposures.** Abbreviations: DIP–diabetes in pregnancy, GDM, gestational diabetes, LGA, large for gestational age.

**Table 3. Comparison of mean maternal blood glucose between birth size categories and preschool age weight categories—ANOVA.**

| | Birth size | | | | | | |
| --- | --- | --- | --- | --- | --- | --- | --- |
| | LGA, N = 131 | AGA, N = 288 | SGA, N = 21 | ANOVA P-value | LGA vs AGA | LGA vs SGA | SGA vs AGA |
| Mean maternal age at booking (years) | 30.79 (6.02) | 30.44 (6.02) | 30.71 (6.61) | 0.854 | | | |
| Mean maternal BMI at booking (kg/m$^2$) | 36.50 (7.07) | 33.79 (8.63) | 32.47 (9.81) | 0.009 | 0.013 | 0.138 | 1.000 |
| Mean FBG at HFDP diagnosis (mmol/L) | 6.58 (1.85) | 6.12 (1.61) | 5.74 (1.77) | 0.020 | 0.041 | 0.131 | 1.000 |
| Mean OGTT 1-hour glucose at HFDP diagnosis (mmol/L) | 11.43 (2.43) | 10.74 (6.38) | 9.51 (2.30) | 0.322 | | | |
| Mean OGTT 2-hour glucose at HFDP diagnosis (mmol/L) | 10.13 (2.43) | 9.45 (2.27) | 9.25 (3.24) | 0.036 | 0.039 | 0.466 | 1.000 |
| Mean third trimester 2-hour postprandial blood glucose (mmol/L) | 6.05 (1.03) | 5.65 (0.79) | 5.32 (0.61) | <0.001 | <0.001 | 0.001 | 0.264 |
| **Overweight and obesity at preschool age** | | | | | | | |
| | Overweight and obese, N = 41 | Normal BMI, N = 88 | Underweight, N = 26 | P-value | | | |
| Mean maternal age at booking (years) | 31.50 (5.02) | 30.66 (6.33) | 29.73 (5.64) | 0.489 | | | |
| Mean maternal BMI at booking (kg/m$^2$) | 37.67 (7.02) | 34.34 (7.97) | 36.16 (10.92) | 0.106 | | | |
| Mean FBG at HFDP diagnosis (mmol/L) | 6.44 (2.05) | 5.92 (1.64) | 6.14 (1.73) | 0.288 | | | |
| Mean OGTT 1-hour glucose at HFDP diagnosis (mmol/L) | 10.70 (2.15) | 10.32 (2.22) | 10.00 (2.05) | 0.464 | | | |
| Mean OGTT 2-hour glucose at HFDP diagnosis (mmol/L) | 9.50(2.50) | 9.10 (2.45) | 8.95 (1.50) | 0.616 | | | |
| Mean third trimester 2-hour postprandial blood glucose (mmol/L) | 5.87 (0.74) | 5.72 (0.85) | 5.67 (1.06) | 0.583 | | | |

*All tests carried out using one-way ANOVA.

Post hoc tests were done with Bonferroni correction and only out when omnibus ANOVA was significant at p = 0.05.

Abbreviations: LGA, large for gestational age, AGA, appropriate for gestational age, SGA, small for gestational age, BMI, body mass Index, FBG, fasting blood glucose, HFDP, hyperglycaemia first detected in pregnancy, OGTT, oral glucose tolerance test.

## Association between maternal blood glucose and offspring weight outcomes at birth and preschool age

In multivariable analysis, maternal 2-hour postprandial glucose in the third trimester was significantly associated with a higher weight z-score at birth (beta 0.20, 95%CI: 0.06–0.34, p = 0.006) and preschool age (beta 0.31, 95%CI: 0.03–0.60, p = 0.032), but not BMI z-score at preschool age (Fig 3). Maternal fasting blood glucose at HFDP diagnosis was also significantly associated with weight z-score at birth (beta 0.10, 95%CI 0.00–0.20, p = 0.046). Notably, maternal BMI at booking of pregnancy was consistently associated with both birthweight z-score and weight z-score at preschool age (Fig 3). However, there were no significant interactions between maternal BMI at booking of pregnancy and the maternal glucose variables (S1-S9 Figs in S1 File).

When weight z-scores were categorized, maternal 2-hour postprandial blood glucose was significantly associated with both SGA at birth (OR 0.41, 95%CI0.17–0.95) and LGA (OR 1.58, 95%CI 1.15–2.16) (S1 Table and S10 Fig in S1 File), after multinomial logistic regression but not with weight or BMI categories at preschool age (S2 Table and S11 Fig in S1 File).

## Discussion

At birth, we found that almost one third (29.6%) of children exposed to HFDP during pregnancy were LGA at birth; 12.2% had macrosomia and 7.5% LBW. Significantly higher

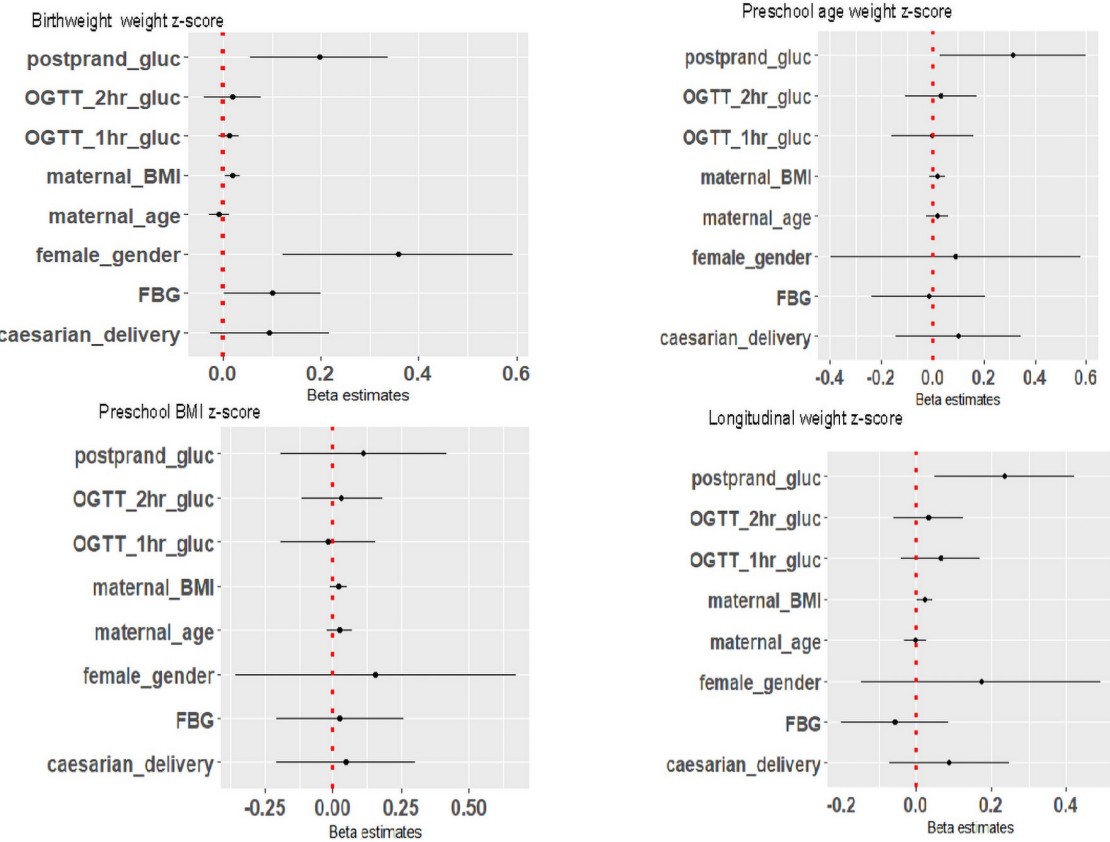

**Fig 3. Association between maternal blood glucose and z-scores for weight at birth and preschool age—Multiple variable linear regression. Abbreviations**: postprandial_gluc–Maternal 2-hour postprandial glucose during the third trimester. OGTT_1hr_gluc– Maternal OGTT 1-hour glucose at HFDP diagnosis. OGTT_2hr_gluc—Maternal OGTT 2-hour glucose at HFDP diagnosis. Maternal_BMI–Maternal Body Mass Index at pregnancy booking. Maternal_age–Maternal age at pregnancy booking. FBG– Maternal fasting blood glucose at HFDP diagnosis. Caesarian_delivery–Caesarian delivery at birth.

proportions of neonates exposed to DIP were either LGA or had macrosomia at birth, compared to GDM-exposed neonates, although there were no significant differences in overweight and obesity at preschool age. We also found that maternal fasting blood glucose at HFDP diagnosis and maternal postprandial blood glucose during the third trimester were both associated with weight z-score at birth.

There are limited data on the prevalence of macrosomia and LGA in South Africa in general and data on the effect of either HFDP or maternal blood glucose levels during pregnancy and offspring weight outcomes in Africa. In one small study from Soweto, South Africa, the prevalence of LGA was 6.4% while only 2.4% of the neonates had macrosomia, with no difference between GDM exposed and unexposed neonates, most likely because of the small size involved [30]. Our findings in a larger cohort show a high prevalence of both LGA and macrosomia at birth and suggest a need for interventions to either prevent GDM or treat more aggressively. LGA and macrosomia are established risk factors for both childhood and adult overweight and obesity [5]. On the other hand, LBW is a risk factor for developmental origins of adult cardiometabolic disease, with findings from several studies showing that infants with LBW have a higher risk for adult high blood pressure, diabetes and dyslipidaemia [6]. In this study, 37.1% of the neonates were either LGA (29.6%) or LBW (7.5%). Apart from interventions to reduce risk of HFDP, and to improve treatment of HFDP, interventions may need to be targeted at

these neonates, at an early age, to reduce risk of both childhood and adulthood overweight and obesity.

We found that both maternal fasting blood glucose at HFDP diagnosis and 2-hour post-prandial glucose during the third trimester were associated with birthweight z-score. Besides, 2-hour postprandial glucose was associated with both SGA and LGA, after adjusting for other glucose indices and maternal BMI at booking of pregnancy. Our findings on the effect of fasting glucose on offspring birthweight are in agreement with published data [12]. However, in contrast to the HAPO study, we found that OGTT 1-hour and 2-hour maternal glucose were not significantly associated with birth size. Instead, we found that postprandial glucose exerted the strongest effect on weight outcomes. While data on the influence of postprandial glucose on offspring weight outcomes are sparse [31], data from trials have shown that tighter control of maternal blood glucose during the pregnancy reduces the risk of adverse perinatal out-comes, including both macrosomia and LGA [32], although data from Africa are scarce. Research is needed to provide evidence of Africa-specific interventions to improve maternal glucose control in pregnancies complicated by HFDP.

We found that neonates exposed to DIP had birthweight z-scores which were 5 times higher than those for neonates exposed to GDM, with corresponding higher proportions of LGA and macrosomia. Research comparing the effects of DIP and GDM on offspring weight outcomes is scarce. Women with DIP do not only have more severe dysglycaemia during the pregnancy but are more likely to have a higher risk of diabetes complications, during and after the preg-nancy [8]. In Africa, the high prevalence of undiagnosed diabetes of 69% [11] raises the possi-bility that some of the women with DIP may have had undiagnosed type 2 diabetes before the pregnancy [33]. In South Africa, 83% of the mothers with DIP progressed to type 2 diabetes within 6 years [10], perhaps partly because a significant proportion may have undiagnosed dia-betes during pregnancy. By extension, offspring of women with DIP are more likely to have been exposed to hyperglycemia for a longer period during the pregnancy, compared to the off-spring of women with GDM. Our findings show a clear need to reduce exposure to untreated DIP in the offspring. One possible solution is to screen for diabetes earlier than the 24-28-week window recommended for GDM. An immediate drawback of the early screening using glucose testing, during the first trimester, for example, is the increased burden in costs and human resources required for this as women who don't have DIP at the initial screening will still need to be screened for GDM later in the pregnancy. One way of overcoming the costly implications of first-trimester glucose testing is by using non-invasive prediction models which use routine clinical data. However, there are currently no prediction models for Africa populations and most of the existing prediction models lack external validity [34]. Recent data from Johannesburg suggest that the use of a dual-threshold fasting plasma glucose, ≥4.5mmol/L to rule out, and 5.1mmol/L to rule in GDM, may result in only 2.4% missed cases of GDM [21]. Using fasting blood glucose only for the diagnosis of HFDP will be less costly than the standard OGTT, and maybe a useful solution in the meantime. The lack of universal screening in South Africa, due to resource limitations is also another limitation as the selective screening currently in use in many provinces in South Africa uses non-validated risk factors and may leave almost half of women with HFDP undiagnosed [16]. A non-invasive screening tool for GDM could be equally usefully as it can be deployed in a universal testing strategy without adding an extra burden to the health system.

At preschool age, we found that more than a quarter (26.5%) of children were either over-weight or obese. This is almost twice as high as the 14% reported from recent South African national surveys [35, 36], and demonstrates a need for intervention in these children. Again, the lack of data from Africa in this area makes comparisons with published literature difficult. Evidence from a retrospective and prospective analysis of 51 505 adults, showed that those

children who were overweight and obese during adolescence experienced the highest increases in BMI during the preschool ages 2–6 years, but not during the school years [4]. Thus, the preschool age may be the critical time during childhood during which susceptibility to adulthood overweight and obesity occurs and when interventions may have the greatest impact.

In this cohort, maternal 2-hour postprandial glucose during the third trimester was associated with weight z-score at preschool age but not preschool age BMI z-scores. In the HAPO follow-up study, there were no significant associations between a GDM diagnosis and child BMI at ages 10–14 years, although the authors found associations between GDM and other measures of adiposity [18]. The 2-hour postprandial blood glucose is a measure of maternal blood glucose control during the third trimester and the possibility that poor glucose control during pregnancy could still exert an effect during preschool years has important clinical implications. More aggressive maternal blood glucose targets during the pregnancy may be required to reduce the risk of both LGA at birth and childhood overweight and obesity in the offspring.

An important observation from our study is the consistent association between maternal BMI and child weight outcomes at birth and preschool age. This may be particularly important in the South African setting where the prevalence of overweight in women was 65% in 2015 [34]. Intervening to reduce pre-conception overweight and obesity may have many multiple benefits as overweight and obesity is also the strongest risk factor for HFDP.

A limitation of our study is the use of routinely collected clinical data for maternal blood glucose values, birthweight and gestational age estimation, where data quality control is not optimum. Conversely, the measurement of maternal blood glucose and birthweight by clinicians unrelated to the study is a strength as clinicians were not aware of the study and these measurements would not have been affected by ascertainment bias. Gestational age was estimated using a combination of LMP and ultrasound but not at the recommended up to 13 6/7 weeks gestation period and this may have resulted in inaccurate estimation. As is usual with all routinely collected data, missing data made it difficult to assess the effect of all variables maternal fasting and postprandial 1-hour glucose during the pregnancy. The loss to follow up in our study was high at preschool age, typical for longitudinal studies in our setting where in-and-out migration is high. Lastly, we did not have access to data from a comparison group of children who were not exposed to HFDP, and future studies should include this group.

## Conclusion

In offspring exposed to HFDP, there is a high prevalence of LGA and macrosomia at birth and overweight and obesity at preschool age, with higher prevalence in offspring exposed to DIP. Poor maternal blood glucose control during the pregnancy, represented by maternal postprandial 2-hour glucose increased risk of both LGA and SGA at birth and was associated with weight z-score at birth and at preschool age. There is need to for earlier screening for HFDP in this setting, to allow earlier detection of DIP and intervention. Improved management of maternal blood glucose during pregnancy is needed to reduce risk of both LGA and SGA at birth and overweight and obesity at preschool age.

## Supporting information

**S1 File. Supplementary tables and figures.**
(PDF)

**S2 File. PRO2D offspring data.**
(XLSX)

**S1 Checklist. STROBE_checklist_v4_pro2d offspring.**
(PDF)

## Acknowledgments

We thank Ms Chantal Stuart (administration) and Ms Siphokazi Khonkwane (data collection) at the Chronic Disease Initiative for Africa for their support during the data collection.

## Author Contributions

**Conceptualization:** Tawanda Chivese, Hetta van Zyl, Naomi S. Levitt, Shane A. Norris.

**Data curation:** Tawanda Chivese, Magret C. Haynes, Hetta van Zyl.

**Formal analysis:** Tawanda Chivese, Una Kyriacos, Naomi S. Levitt, Shane A. Norris.

**Funding acquisition:** Tawanda Chivese, Naomi S. Levitt.

**Investigation:** Tawanda Chivese, Magret C. Haynes, Naomi S. Levitt, Shane A. Norris.

**Methodology:** Tawanda Chivese, Naomi S. Levitt, Shane A. Norris.

**Project administration:** Tawanda Chivese.

**Resources:** Naomi S. Levitt.

**Software:** Tawanda Chivese.

**Supervision:** Naomi S. Levitt, Shane A. Norris.

**Validation:** Tawanda Chivese.

**Writing – original draft:** Tawanda Chivese.

**Writing – review & editing:** Tawanda Chivese, Magret C. Haynes, Hetta van Zyl, Una Kyriacos, Naomi S. Levitt, Shane A. Norris.

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
