## [Decision Letter · Decision Letter 0]

12 Aug 2021

PONE-D-21-11542

The influence of maternal blood glucose during pregnancy on weight outcomes at birth and preschool age in offspring exposed to hyperglycemia first detected during pregnancy, in a South African cohort

PLOS ONE

Dear Dr. Chivese,

Thank you for submitting your manuscript to PLOS ONE. After careful consideration, we feel that it has merit but does not fully meet PLOS ONE’s publication criteria as it currently stands. Therefore, we invite you to submit a revised version of the manuscript that addresses all of the points raised by the reviewers during the review process.

We look forward to receiving your revised manuscript.

Kind regards,

Pratibha V. Nerurkar, Ph.D

Academic Editor

PLOS ONE

Journal Requirements:

2. Thank you for stating in the text of your manuscript "mothers gave informed consent for their children to participate and each child assented before they had their weight and height measured." Please state what type of consent you obtained (for instance, written or verbal, and if verbal, how it was documented and witnessed). Please also add all of this information to your ethics statement in the online submission form.

Reviewers' comments:

Reviewer's Responses to Questions

**Comments to the Author**

1. Is the manuscript technically sound, and do the data support the conclusions?

Reviewer #1: Yes

Reviewer #2: Partly

2. Has the statistical analysis been performed appropriately and rigorously? 

Reviewer #1: Yes

Reviewer #2: No

3. Have the authors made all data underlying the findings in their manuscript fully available?

Reviewer #1: Yes

Reviewer #2: Yes

4. Is the manuscript presented in an intelligible fashion and written in standard English?

Reviewer #1: Yes

Reviewer #2: Yes

5. Review Comments to the Author

Reviewer #1: Overall, this is a worthy study as it provides new information of an under-represented population. The authors should be commended on their long-term follow-up of their cohort, which is a difficult task.

A few comments/considerations before publication

Background:

Please describe clinical significance and meaning of DIP vs. GDM. This is not a usual classification in the United States, and for the majority of the paper, I thought DIP implied there was Type 2 DM present. In the conclusion it was clear Type 2 is a different entity. Please provide further definition and clinical significance.

Methods and Study Design:

Due to the long-term follow up of this study, performed many years ago, I assume there was no healthy controls that were followed? If there were, consider comparing the diabetic cohort to a healthy cohort. Otherwise, please provide the reader with an understanding of what baseline rates of macrosomia, LBW, LGA, SGA are in your community. It is difficult for the reader to ascertain the magnitude of these outcomes compared to healthy parturients.

Statistical Analysis:

Consider performing (or displaying if already done) an interaction analysis on the modifying effect of maternal BMI on Glucose and neonatal/preschool weight. The authors mention that this was associated, but I do not see the data displayed.

There are some typos - 'Casearian (table 1, page 14) and marternal (in table 3, page 17)

Reviewer #2: Overall, this is an interesting paper looking at the relationship between maternal blood glucose during pregnancy and weight outcomes at birth with those at preschool age. The methodology appears OK and the authors correctly used ultrasound estimation in addition to calculation from the first day of the last menstrual period to estimate gestational age. However, it is not clear if the ultrasounds were done in a routine fashion up to 13 6/7 weeks gestation. Also, did any pregnancies result from in vitro fertilization, and if so, how was gestational age handled? My only concern in this otherwise nicely written manuscript is the use of logistic versus log-binomial regression to compute effect estimates. Logistic regression is prone to non-collapsibility bias and also may yield inflated risk estimates.

6. PLOS authors have the option to publish the peer review history of their article (what does this mean?). If published, this will include your full peer review and any attached files.

Reviewer #1: No

Reviewer #2: No

---

## [Author Response · Author response to Decision Letter 0]

27 Aug 2021

PONE-D-21-11542

The influence of maternal blood glucose during pregnancy on weight outcomes at birth and preschool age in offspring exposed to hyperglycemia first detected during pregnancy, in a South African cohort

PLOS ONE

Dear Editor

Thank you for your feedback on our submitted manuscript to PLOS ONE. We are grateful to both the journal and the reviewers for taking their time to help strengthen this paper. We have made the necessary changes to the manuscript as detailed in this rebuttal.

We look forward to your kind response

Kind regards,

Tawanda Chivese

Corresponding author

Journal Requirements:

Thank you, we have addressed this.

2. Thank you for stating in the text of your manuscript "mothers gave informed consent for their children to participate and each child assented before they had their weight and height measured." Please state what type of consent you obtained (for instance, written or verbal, and if verbal, how it was documented and witnessed). Please also add all of this information to your ethics statement in the online submission form.

Thank you. The mothers gave written informed consent. This has been updated in the manuscript and the online submission form

Thank you, we have uploaded the data as Supporting Information

Thank you, we have done so

Reviewers' comments:

Reviewer #1: Overall, this is a worthy study as it provides new information of an under-represented population. The authors should be commended on their long-term follow-up of their cohort, which is a difficult task.

A few comments/considerations before publication

Background:

Please describe clinical significance and meaning of DIP vs. GDM. This is not a usual classification in the United States, and for the majority of the paper, I thought DIP implied there was Type 2 DM present. In the conclusion it was clear Type 2 is a different entity. Please provide further definition and clinical significance.

Thank you for this important input. We have added more detail in the background to describe the clinical significance of the distinction of DIP vs. GDM. In brief, we classified DIP as diabetes mellitus first recognized during pregnancy per WHO 2013 criteria. Because of the difficulties involved in diagnosing diabetes in pregnancy, women who present with blood glucose levels within the levels of diabetes outside pregnancy are not classified as type 2 diabetes, as there are insufficient data about whether their blood glucose levels normalize or not after the pregnancy. This is the group of women who we classified as DIP. In Africa, because of the high prevalence of undiagnosed diabetes, a proportion of women in this category probably have preexisting diabetes but that proportion is unknown due to a lack of research in this area.

Methods and Study Design:

Due to the long-term follow up of this study, performed many years ago, I assume there was no healthy controls that were followed? If there were, consider comparing the diabetic cohort to a healthy cohort. Otherwise, please provide the reader with an understanding of what baseline rates of macrosomia, LBW, LGA, SGA are in your community. It is difficult for the reader to ascertain the magnitude of these outcomes compared to healthy parturients.

Thank you, studies on prevalence of LGA and macrosomia are scarce from South Africa. We have included data from the one study we found. We also included data from one nationally representative study which showed that the prevalence of childhood overweight and obesity was 14% in children aged South Africa. This is also discussed in the limitations section of the discussion.

Statistical Analysis:

Consider performing (or displaying if already done) an interaction analysis on the modifying effect of maternal BMI on Glucose and neonatal/preschool weight. The authors mention that this was associated, but I do not see the data displayed.

Thank you, we have carried out the interaction analysis and this is included as supplementary Figures. However, the interaction term was not statistically significant in the analysis and therefore we did not explore this further.

There are some typos - 'Casearian (table 1, page 14) and marternal (in table 3, page 17)

Thank you, these have been addressed. We have also checked the whole manuscript for typos and grammatical errors.

Reviewer #2: Overall, this is an interesting paper looking at the relationship between maternal blood glucose during pregnancy and weight outcomes at birth with those at preschool age. The methodology appears OK and the authors correctly used ultrasound estimation in addition to calculation from the first day of the last menstrual period to estimate gestational age. However, it is not clear if the ultrasounds were done in a routine fashion up to 13 6/7 weeks gestation. 

Thank you. The policy in the Western province of South Africa at the time of the index pregnancies recommended a routine ultrasound examination at 18-23 weeks rather than up to 13 6/7 weeks gestation for low-risk pregnancies. We have explained this in the methods section and added this as a limitation. 

Also, did any pregnancies result from in vitro fertilization, and if so, how was gestational age handled? 

Thank you, there were no pregnancies which had resulted from in-vitro fertilization, and therefore this was not discussed.

My only concern in this otherwise nicely written manuscript is the use of logistic versus log-binomial regression to compute effect estimates. Logistic regression is prone to non-collapsibility bias and also may yield inflated risk estimates.

Thank you, we have reported exponentiated coefficients from the linear regression. However, for the analysis of categorised LGA and overweight and obesity at preschool age, log-binomial regression failed to converge for LGA while it converged for overweight and obesity at preschool age. We carried out Poisson regression for LGA and log-binomial regression for preschool overweight and obesity and presented both outputs as supplementary analysis.

---

## [Decision Letter · Decision Letter 1]

8 Oct 2021

The influence of maternal blood glucose during pregnancy on weight outcomes at birth and preschool age in offspring exposed to hyperglycemia first detected during pregnancy, in a South African cohort

PONE-D-21-11542R1

Dear Dr. Chivese,

We’re pleased to inform you that your manuscript has been judged scientifically suitable for publication and will be formally accepted for publication once it meets all outstanding technical requirements.

Kind regards,

Pratibha V. Nerurkar, Ph.D

Academic Editor

PLOS ONE

Additional Editor Comments (optional):

Reviewers' comments:

Reviewer's Responses to Questions

**Comments to the Author**

1. If the authors have adequately addressed your comments raised in a previous round of review and you feel that this manuscript is now acceptable for publication, you may indicate that here to bypass the “Comments to the Author” section, enter your conflict of interest statement in the “Confidential to Editor” section, and submit your "Accept" recommendation.

Reviewer #1: All comments have been addressed

Reviewer #2: All comments have been addressed

2. Is the manuscript technically sound, and do the data support the conclusions?

Reviewer #1: (No Response)

Reviewer #2: (No Response)

3. Has the statistical analysis been performed appropriately and rigorously? 

Reviewer #1: (No Response)

Reviewer #2: (No Response)

4. Have the authors made all data underlying the findings in their manuscript fully available?

Reviewer #1: (No Response)

Reviewer #2: (No Response)

5. Is the manuscript presented in an intelligible fashion and written in standard English?

Reviewer #1: (No Response)

Reviewer #2: (No Response)

6. Review Comments to the Author

Reviewer #1: (No Response)

Reviewer #2: (No Response)

7. PLOS authors have the option to publish the peer review history of their article (what does this mean?). If published, this will include your full peer review and any attached files.

Reviewer #1: No

Reviewer #2: No

---

## [Editor Report · Acceptance letter]

13 Oct 2021

PONE-D-21-11542R1 

The influence of maternal blood glucose during pregnancy on weight outcomes at birth and preschool age in offspring exposed to hyperglycemia first detected during pregnancy, in a South African cohort 

Dear Dr. Chivese:

I'm pleased to inform you that your manuscript has been deemed suitable for publication in PLOS ONE. Congratulations! Your manuscript is now with our production department. 

Kind regards, 

on behalf of

Dr. Pratibha V. Nerurkar 

Academic Editor

PLOS ONE